# Barriers and facilitators to the conduct of critical care research in low and lower-middle income countries: A scoping review

**Bharath Kumar Tirupakuzhi Vijayaraghavan**[1]*, **Ena Gupta**[2], **Nagarajan Ramakrishnan**[1], **Abi Beane**[3], **Rashan Haniffa**[3,4], **Nazir Lone**[5], **Nicolette de Keizer**[6], **Neill K. J. Adhikari**[7]

1 Department of Critical Care Medicine, Apollo Hospitals, Chennai, India, 2 Department of Pulmonary and Critical Care Medicine, Einstein Health Network, Philadelphia, Pennsylvania, United States of America, 3 Mahidol-Oxford Tropical Research Unit, Bangkok, Thailand, 4 Department of Anaesthesia and Critical Care Medicine, University College London, London, United Kingdom, 5 Usher Institute, University of Edinburgh, Edinburgh, United Kingdom, 6 Department of Medical Informatics, Amsterdam Public Health Research Institute, Amsterdam UMC, University of Amsterdam, Amsterdam, The Netherlands, 7 Interdepartmental Division of Critical Care Medicine, Department of Critical Care Medicine, Sunnybrook Health Sciences Centre, University of Toronto, Toronto, Canada

* bharath@icuconsultants.com

## Abstract

### Background

Improvements in health-related outcomes for critically ill adults in low and lower-middle income countries need systematic investments in research capacity and infrastructure. High-quality research has been shown to strengthen health systems; yet, research contributions from these regions remain negligible or absent. We undertook a scoping review to describe barriers and facilitators for the conduct of critical care research.

### Methods

We searched MEDLINE and EMBASE up to December 2021 using a strategy that combined keyword and controlled vocabulary terms. We included original studies that reported on barriers or facilitators to the conduct of critical care research in these settings. Two reviewers independently reviewed titles and abstracts, and where necessary, the full-text to select eligible studies. For each study, reviewers independently extracted data using a standardized data extraction form. Barriers and facilitators were classified along the lines of a previous review and based on additional themes that emerged. Study quality was assessed using appropriate tools.

### Results

We identified 2693 citations, evaluated 49 studies and identified 6 for inclusion. Of the included studies, four were qualitative, one was a cross-sectional survey and one was reported as an 'analysis'. The total number of participants ranged from 20–100 and included physicians, nurses, allied healthcare workers and researchers. Barriers identified included limited funding, poor institutional & national investment, inadequate access to mentors, absence of training in research methods, limited research support staff, and absence of

**Data Availability Statement:** All data are available in the manuscript file and the supplementary files.

**Funding:** Authors RH and AB were co-applicants on the grant. Wellcome Trust, U.K. (grant number WT215522/Z19/Z). https://wellcome.org/ The funder had no role in the design, conduct, analysis of this scoping review or in the decision to submit for publication

**Competing interests:** The authors have declared that no competing interests exist.

statistical support. Our review identified potential solutions such as developing a mentorship network, streamlining of regulatory processes, implementing a centralized institutional research agenda, developing a core-outcome dataset and enhancing access to low-cost technology.

## Conclusion

Our scoping review highlights important barriers to the conduct of critical care research in low and lower-middle income countries, identifies potential solutions, and informs researchers, policymakers and governments on the steps necessary for strengthening research systems.

## Background

Over 75% of the global population resides in low or lower-middle income settings, as defined by the World Bank [1], and faces an enormous burden of communicable and non-communicable disease. Improvements in health-related outcomes in these regions requires systematic investments and focus on health-related infrastructure, public health capacity, training, general sanitation and hygiene, and poverty alleviation. In addition, and equally crucial, are investments in research capacity and infrastructure. While the health-related problems in these regions are often unique, locally led research solutions are either inadequate or non-existent [2]. Apart from describing epidemiology or developing diagnostic and prognostic tools or testing interventions, high quality research has been shown to strengthen health systems, especially in countries and settings where such systems are typically fragile [3]. And yet, across disciplines and specialities, the research contributions from low and lower-middle income settings are negligible or absent [4–6]. Specific to critical illness, the epidemiology in low and lower-middle income countries (LMICs) is distinct from high income countries (HIC) in several ways: diseases that bring patients into intensive care units (ICUs) (e.g. tropical infections, toxicology, snake and scorpion bites) [7–9], burden of antimicrobial resistance [10, 11], resources and expertise available for treatment, organization and provision of critical care as a service, quality of care provided, and outcomes from an episode of critical illness [12] In the absence of context-specific information, most ICUs in LMICs are forced to appraise and apply evidence generated from HIC settings. This situation creates gaps in evidence availability and in knowledge translation. The postulated reasons for this absence of context-relevant data include a heavy clinical burden, lack of research infrastructure and training in relevant skills, absence of funding, regulatory barriers and ambiguities, and the low priority accorded by governments and healthcare systems to research in these regions [13].

A previous systematic review published in 2018 focused on barriers for the conduct of clinical trials in developing countries [14]. However, as this review was focused on trials, rather than research using broader methodologies, and did not specifically focus on the critical care setting, we aimed to perform a review of the literature to describe the barriers and opportunities for the conduct of research in critical care settings of LMICs.

## Methods

### Search strategy and eligibility criteria for studies

With the help of a librarian, we searched Ovid versions of MEDLINE and EMBASE for all relevant publications from inception to December 2021. We used a strategy that combined

multiple keyword terms and controlled vocabulary search terms covering 'critical care' and 'barriers and facilitators' and 'research'. The detailed search strategy is provided in the S1 Appendix. Additionally, we screened the reference lists of all included articles. Based on the research teams' language knowledge and the lack of resources to include a translator, the search was restricted to English-language publications.

We included original studies that reported on barriers or facilitators to the conduct of critical care research in LMICs. For this review, we included studies that used qualitative or quantitative approaches, or a mixture. We excluded editorials, commentaries, letters to editor and other non-peer reviewed work. For the purposes of this review, we included countries that have been classified as low or lower-middle income as per the World Bank country and lending groups classification [1]. Since the definition of critical care is variable, we included all studies where authors have identified the population as critically ill.

We used a modified PICO (Population, Intervention or Exposure, Comparator, Outcome) approach for study selection. Mapped to the Population-Concept-Context (PCC) framework designed by the Joanna Briggs Institute for Scoping Reviews, the corresponding PCC would be:

Population: critically ill patients of any age
Concept: barriers and facilitators to the conduct of critical care research
Context: Low and Lower-middle income countries

## Study selection, data extraction and quality assessment

Two reviewers (BKTV and EG), both having critical care clinical and research experience in a LMIC setting, independently reviewed titles and abstracts, and where necessary, the full-text of identified articles to select eligible studies (as defined above). For each eligible study, the same two reviewers independently extracted data on barriers and facilitators. A standard data extraction form was designed, and pilot tested prior to extraction. For all studies, we extracted data on study country or region, design, population of interest and barriers and facilitators to research that are identified. Barriers and facilitators were classified broadly along the lines of the previous review [14] and based on additional themes that emerged. Disagreements, if any, were resolved in consultation with a third reviewer (NKJA).

Study quality was assessed using the tools developed by the CLARITY research group for all non-randomized designs [15], and criteria proposed by Kuper for qualitative studies [16]. Study quality was adjudicated independently by the same two reviewers and disagreements resolved as above.

## Data analysis

We described the individual study settings, populations studied, design and key observations. In addition, we categorized by themes the barriers and facilitators identified across all the included studies.

## Ethics, registration and reporting

Based on the study design, we did not seek ethics committee approval. The protocol was registered as a preprint on *Open Science Forum (*doi:10.17605/OSF.IO/9UQNS) prior to analysis [17]. The review is being reported as per the Preferred Reporting Items for Systematic Reviews and Meta-analyses extension for Scoping Reviews (PRISMA-ScR) checklist (S4 Appendix) [18].

**Deviations from protocol.** While our original protocol specified inclusion of only original articles and articles restricted to critical care, we had to make exceptions to these

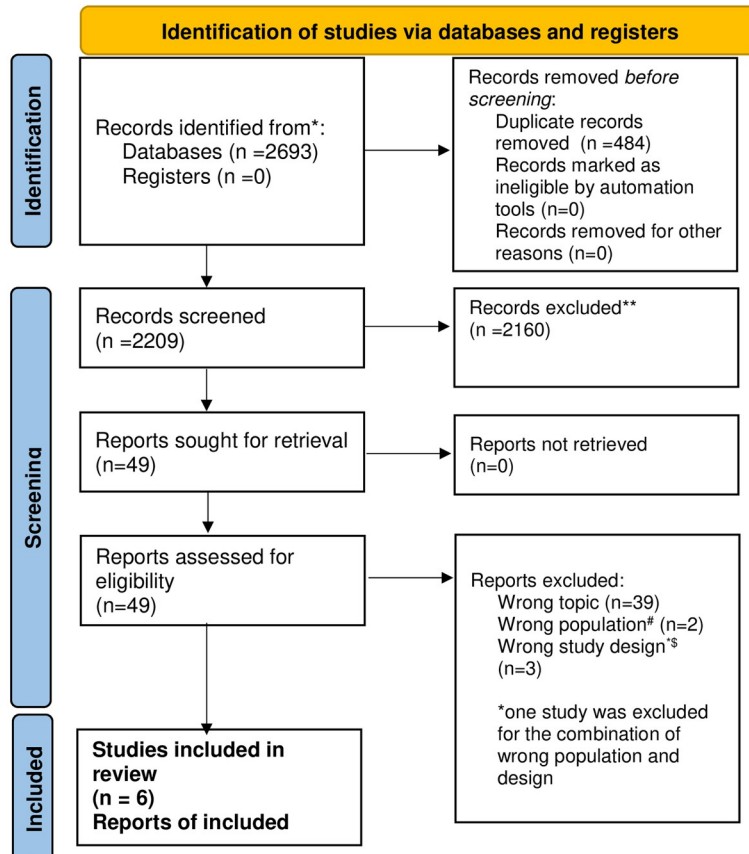

**Fig 1. PRISMA 2020 flow diagram for new systematic reviews which included searches of databases and registers only.**

prespecified rules in two instances. Our search yielded only 6 papers of relevance and we were keen to maximise information and learning. Hence, we shifted our strategy from a restrictive approach to including these studies in order to maximise insights on barriers and facilitators to the conduct of research in LMICs. We believe this approach is in alignment with the broad objectives and definitions of scoping reviews.

## Results

### Study flow

We identified 2693 citations from the search of electronic records (Ovid MEDLINE and EMBASE) and a review of the bibliography of the included studies and relevant review articles. We evaluated 49 studies in detail and identified 6 studies for inclusion (Fig 1). The two reviewers (BKTV and EG) achieved complete agreement on the included studies.

### Study characteristics

We present the characteristics of the 6 included studies in Table 1. Of the included studies, 4 were qualitative [19–22], 1 was a cross-sectional survey [23] and the study by Aluisio and

**Table 1. Characteristics of the included studies.**

| Author, Location/s | Year of publication | Subjects (n) | Type/ Methods | Focus of study | Additional details |
|---|---|---|---|---|---|
| Ahmed [21] Africa and Latin America | 2020 | Participants from Africa and Latin America (n = 21 in first year and 40 in second year) | Qualitative–focus group discussions | Barriers and strategies | Focus group discussions at the annual American Society of Tropical Medicine and Hygiene conference, led by researchers who had repatriated to home countries, with participants keen on taking a similar route. |
| Aluisio [24] Low and lower-middle income countries | 2019 | NA | Analysis | Challenges and opportunities | Analysis of the challenges and opportunities for clinical emergency care research in low and lower-middle income countries. |
| Franzen [19] Ethiopia, Cameroon and Sri Lanka | 2017 | Participants from Ethiopia, Cameroon and Sri Lanka (n = 57) | Qualitative–mixed methods (interviews, focus group discussions and process mapping) | Barriers and enablers | Qualitative evaluation of barriers and enablers to health research capacity from Ethiopia, Cameroon, and Sri Lanka. |
| Johnson [22] Colombia | 2021 | Participants from three centres in Colombia and the coordinating centre for the registry (n = 20) | Qualitative mixed methods (interviews and ethnography) | Barriers and enablers | Qualitative evaluation of stakeholders from 3 centres in Colombia focused on the challenges to implementing a neurotrauma registry |
| Sawe [20] Tanzania | 2020 | Participants from 5 centres in Tanzania (n = 49) | Qualitative—focus group discussions | Barriers and facilitators | Qualitative study examining barriers and facilitators to the collection of high-quality data for trauma patients with the intention of exploring opportunities and challenges for establishment of a trauma registry |
| Andre-von Arnim [23] Southeast Asia, Africa, Latin America and Eastern Europe | 2017 | Participants from Southeast Asia, Africa, Latin America and Eastern Europe (n = 47; predominantly from Latin America) | Quantitative -survey | Challenges and priorities | Survey of clinician scientists from LMICs regarding priorities and challenges for pediatric critical care research |

LMIC, low-and lower middle income country.

colleagues was described to be an 'analysis' [24]. All included studies were published as full articles. The study by Ahmed [21] included participants from Africa and Latin America; the study by Franzen [19] included participants from Ethiopia, Cameroon and Sri Lanka. The study by Johnson and colleagues [22] included participants from Colombia and Sawe's study [20] included participants from Tanzania. The study by von Arnim [23] included participants from Southeast Asia, Latin America, Africa and Eastern Europe. The publication by Aluisio [24] was reported as an 'analysis' paper and the population of interest was broadly low and lower-middle income countries. The total number of participants ranged from 20–100 across the different studies and included physicians, nurses, allied healthcare workers and researchers.

## Barriers and facilitators

Table 2 reports on the common barriers identified from the 6 studies and Table 3 provides details of the facilitators to research or the proposed solutions emerging from these studies. Barriers and facilitators were broadly classified under 7 themes–Finance, Human capacity/factors, Ethics, Governance and Regulatory, Research Environment, Operational, Competing demands and Others.

Common barriers include the limited funding available for research and for conference travel in LMICs, the poor institutional and national investment in research, inadequate access

**Table 2. Barriers for the conduct of critical care research in LMICs.**

| Theme | Sub-theme | References |
|---|---|---|
| Financial | Limited local funding for research and for projects in LMICs and limited national investment in research in general<br>Forced to depend on international grant funding with low chances of success<br>Limited funding for travel to International conferences which in turn reduces global peer support and networking | **Ahmed [21], Aluisio [23], Franzen [19] and Andre-von Arnim [23]** |
| Human capacity/ factors | Lack of access to mentors<br>Lack of awareness among researchers of local research priorities<br>Limited research support staff<br>Limited access to Statistical support<br>Fewer trained researchers per capita in LMICs and low confidence among researchers<br>Attitudes with respect to research related documentation among researchers | **Ahmed [21], Andre-von Arnim [23], Franzen [19], Johnson [22] and Aluisio [24]**<br><br>**Sawe [20]** |
| Ethical, governance and regulatory issues | Issues related to Ethics Committee/Institutional review board<br>Less regulatory infrastructure with weak systems and limited guidance and oversight<br>Paucity of training in ethical frameworks for personnel<br>Higher prevalence of vulnerable patients which makes ethical issues more challenging<br>Bureaucratic organizations and centralized hierarchies in academic institutions<br>Multiple permissions and approvals needed for research<br>Financial regulations inhibit purchasing of materials | **Andre-von Arnim [23], Franzen [19], Johnson [22], Aluisio [24]** |
| Research environment and infrastructure | Limited infrastructure and equipment in laboratories including access to materials and poor internet connectivity<br>Lack of or limited research job opportunities<br>Access to scientific material (journals, databases, etc.) is limited<br>Limited training opportunities<br>Lack of dedicated research time<br>Lack of autonomy in research projects<br>Difficulty with publications, both in accessing published literature and in getting published<br>Absence of reliable medical records<br>Limited teamwork, local networking and collaborative spirit<br>Inconsistent documentation and archiving systems | **Ahmed [21], Aluisio [24], Andre-von Arnim [23], Franzen [19]**<br>**Franzen [19]**<br>**Franzen [19]**<br>**Sawe [20]** |
| Operational barriers | Difficulties with patient recruitment<br>Trial operations are complex and start-up stages are cumbersome<br>Burden of data collection when high seen as a barrier | **Andre-von Arnim [23]**<br>**Franzen [19]**<br>**Johnson [22]** |
| Competing demands | High clinical burden<br>Complexity and severity of diseases | **Johnson [22], Andre-von Arnim [23], Aluisio [24]** |
| Others | Lower access to technology and lower comfort levels with use of technology<br>Poor internet connectivity and information technology support<br>Unfeasible outcomes in studies of emergency and critical care, e.g. 90 day mortality | **Johnson [22], Aluisio [24]** |

**Table 3. Facilitators and proposed solutions for the conduct of critical care research in LMICs.**

| Theme | Facilitators/Solutions proposed | References |
|---|---|---|
| Finance | Greater national and institutional investment<br>Public-private partnerships in LMICs<br>Collaborative effort between local researchers to seek funding<br>Partnerships between HIC and LLMICs and joint applications for funding | **Ahmed [21], Aluisio [24], Franzen [19], Sawe [20]** |
| Human capacity/ factors | Develop a local mentorship network that can actively promote junior colleagues and facilitate access to mentors from HICs<br>Foster networking and knowledge sharing between local researchers<br>Staying connected with local research priorities by engaging with national and regional professional organizations<br>Motivated and driven clinicians and researchers | **Ahmed [21], Johnson [22], Franzen [19], Andre-von Arnim [23]** |
| Ethical, Governance and Regulatory | Development of research ethics boards in LMICs which needs investment from local institutions and partnerships between institutions in these regions<br>Training in research ethics and trial design<br>Greater resources for Ethics Committees and legal backing<br>Greater research-policy interaction and engagement<br>Streamlined IRB review | **Aluisio [24], Franzen [19], Andre-von Arnim [23]** |
| Research Environment | Collaboration with HIC partners for sharing research resources (e.g. lab resources)<br>To overcome the problem of limited job opportunities, researchers to consider framing proposals around local priorities and develop multi-disciplinary skillsets<br>Opportunities for career progression for researchers and other incentives<br>Institutional support for accessing journals and databases<br>Training researchers in methods starting from medical school<br>Development of a centralised institutional research agenda<br>Training in grant writing skills and work-based training<br>Inculcating a research culture at the institutional level and stakeholder engagement<br>Events to enable networking e.g. opportunity to participate at conferences<br>Improved medical records<br>Local support provided by hospital administration | **Ahmed [21], Johnson [22], Aluisio [24], Franzen [19], Andre-von Arnim [23]** |
| Operational | Facility commitment to standardizing care- which will promote research | **Sawe [20]** |
| Competing demands | Protected research time | **Andre-von Arnim [23]** |
| Others | Enhancing access to low-cost technology including mobile phones, tablet-devices for data collection and telemedicine<br>The presence of an easy-to-use online data collection tool and flexible data collection platform<br>Outcomes for studies should be based on context and available resources<br>A minimum core dataset to be developed in LMICs for specific disciplines<br>Addition of structure and process metrics in studies along with clinical and patient data<br>Providing rationale and context to stakeholders about the research being undertaken<br>Ability of a project to serve future and long-term needs seen as a facilitator | **Johnson [22], Aluisio [24]** |

to mentors, a lack of awareness of local research priorities, absence of training in research methods, limited research support staff, and absence of statistical support. Additional barriers related to ethical and governance systems including the need for multiple approvals, and the weak regulatory frameworks in place in these regions.

## Quality of included studies

We were able to adjudicate on the quality of all studies with the exception of the publication by Aluisio et al. Given the different study designs and tools, we did not provide a global quality rating, but have made our assessments available as an appendix (S2 Appendix- Ahmed et al., Franzen et al. Johnson et al and Sawe et al. and S3 Appendix- von Arnim et al.). Broadly, studies were of moderate quality, in that each one partially satisfied quality requirements.

## Discussion

Our scoping review sheds light on the key barriers and facilitators to the conduct of critical care research in low and lower-middle income countries. The absence of funding, the poor national and institutional investment in research, absence of mentors, the limited research support infrastructure, unreliable medical records, lack of research methods training, ethical and regulatory issues, and insufficient statistical support are key recurring themes. In addition, the high clinical burden, complex trial operations and the choice of outcomes in acute care research also emerge as barriers.

In a previous review, Alemayehu and colleagues examined the barriers for the conduct of clinical trials in developing countries and reviewed the published literature between the years 1995–2015 [14]. Their review broadly identified similar barriers related to funding, ethical and regulatory system obstacles, absence of research infrastructure, logistics and the competing demands on researchers. In addition to these, our study identified barriers related to the choice of outcomes in emergency/critical care research, the absence or the poor quality of medical records and documentation hindering research, and bureaucratic hurdles and the need for multiple permissions. In contrast to their review, our search was broader and extended from inception of databases to March 2021; we focused specifically on critical care research, but broadly on barriers and facilitators to all types of research (and study designs) and did not limit ourselves to clinical trials. Trials are, by definition, complex and large undertakings and perhaps understandably harder to design and execute in countries and regions with an absent or a nascent research infrastructure and culture.

In 2004, the World Health Organization, in its "World Report On Knowledge for Better Health", emphasized on the need for research as a fundamental component of solutions aimed at improving health in all countries [25]. Despite this, most clinical research continues to be funded, designed, and conducted only in HIC settings. For instance, in an analysis by Thiers and colleagues of country-specific data on trial participation, 66% of all trial sites were from 5 countries in North America, Western Europe and Oceania [26]. While the paper highlighted several encouraging trends suggesting improvement in trial participation from non-HIC settings, gaps remain large.

A 2017 report from the Academy of Medical Sciences, U.K. on "Strengthening clinical research capacity in low-and middle-income countries" [27] identified similar gaps and barriers as highlighted by our review. In addition, the report identified opportunities for strengthening clinical research, including suggestions for formalising career pathways, promoting clinical research early in the professional training of clinicians, and in connecting young scientists with the various stakeholders involved in clinical research in their respective countries and regions.

## Implications for practice and research

Our scoping review throws light on some of the persistent challenges to the strengthening of research systems in LMICs. It identifies deep-rooted issues that have plagued healthcare systems in resource-constrained settings. Yet, the review also calls attention to potential solutions and opportunities, several of which are immediately feasible and implementable. Some of these relate to greater collaboration between HIC and LMIC researchers, including joint funding applications, development of a strong mentorship network within LMICs as well as between LMIC and HIC researchers, and the sharing of resources (equipment, technology etc.) between these regions [28, 29]. Additional solutions include training clinicians in research methods as well as in grant-writing and manuscript writing [30, 31], improving the quality of medical documentation and record keeping, and enhancing access to low-cost ubiquitous technology such as mobile phones and tablet-devices for easier data collection and entry. Other solutions specific to critical care research include the development of core dataset for specific disciplines and choosing outcomes that are both relevant, context-specific and feasible [32, 33].

Encouragingly, several newer models are addressing some of the above challenges- the development of critical care registries in Asia and Africa focused on evaluating case-mix and outcomes from critical illness with harmonized data collection tools [34, 35], the embedding of the Randomized Embedded Multifactorial Adaptive Platform Trial for Community Acquired Pneumonia (REMAP-CAP) within these registries [36], the recently published international collaborative trial comparing two doses of corticosteroids in severe COVID-19 with nearly 40% of trial participants enrolled from India [37] the World Health Organization led pragmatic SOLIDARITY trial [38] are some important examples.

## Strengths and limitations

Our review has several important strengths. We used robust scoping review methods: two researchers searched the databases and extracted data independently; the data abstraction form was piloted; and we did not limit our search to a narrow time period or to a specific research design. We extracted key concepts from the included studies and categorized them into practical themes. Our results inform researchers, policy makers and governments in LMICs on the steps necessary for strengthening research systems in their respective countries.

Our review also has several key limitations. We restricted our search to the two databases MEDLINE and EMBASE as we only found 6 studies that could be included from these two large and popular databases with robust indexing methods. In discussion between the authors, it was strongly felt that the additional effort needed to search other databases would be disproportionate to the likely success in finding additional literature. The 'effort to yield' ratio was thought to be low and hence we made the decision to stop with the two largest databases. Included studies were of 'moderate' quality and we were unable to provide a global rating for each study. While several of the identified themes overlap with previously identified concepts, our review highlights additional barriers and facilitators, several of which are readily addressable.

## Conclusion

Our scoping review highlights important and persistent barriers to the conduct of critical care research in LMICs, identifies potential solutions, and informs researchers, policy-makers and governments on the steps necessary for strengthening research systems. While there have been recent encouraging examples that address some of these challenges, broader, multifaceted and systematic strategies with short and longer term goals are essential from Ministries of Health,

Public health agencies and other key stakeholders to addressing the deep-rooted problems that have plagued research in LMICs.

## Supporting information

**S1 Appendix. Search strategy.**
(DOCX)

**S2 Appendix. Study quality assessment (for references [20–23]).**
(DOCX)

**S3 Appendix. Study quality assessment (for reference [24]).**
(PDF)

**S4 Appendix. PRISMA scoping review checklist.**
(PDF)

## Acknowledgments

We would like to acknowledge the assistance of Mr. Henry Lam and Miss. Taylor Moore, Senior Librarians at Sunnybrook Health Sciences, Toronto, Canada for their assistance with the development and implementation of the search strategy and Dr Lakshmi Ranganathan, Clinical Research Manager at Chennai Critical Care Consultants Pvt. Limited, Chennai, India for her assistance in manuscript formatting.

## Author Contributions

**Conceptualization:** Bharath Kumar Tirupakuzhi Vijayaraghavan, Ena Gupta, Abi Beane, Rashan Haniffa, Nazir Lone, Nicolette de Keizer, Neill K. J. Adhikari.

**Data curation:** Bharath Kumar Tirupakuzhi Vijayaraghavan.

**Formal analysis:** Bharath Kumar Tirupakuzhi Vijayaraghavan.

**Funding acquisition:** Abi Beane, Rashan Haniffa.

**Investigation:** Bharath Kumar Tirupakuzhi Vijayaraghavan, Ena Gupta, Nagarajan Ramakrishnan, Abi Beane, Rashan Haniffa, Nazir Lone, Nicolette de Keizer, Neill K. J. Adhikari.

**Methodology:** Bharath Kumar Tirupakuzhi Vijayaraghavan, Ena Gupta, Abi Beane, Rashan Haniffa, Nazir Lone, Nicolette de Keizer, Neill K. J. Adhikari.

**Project administration:** Bharath Kumar Tirupakuzhi Vijayaraghavan, Ena Gupta, Nagarajan Ramakrishnan, Abi Beane, Rashan Haniffa, Nicolette de Keizer, Neill K. J. Adhikari.

**Supervision:** Nagarajan Ramakrishnan, Abi Beane, Rashan Haniffa, Nazir Lone, Nicolette de Keizer, Neill K. J. Adhikari.

**Writing – original draft:** Bharath Kumar Tirupakuzhi Vijayaraghavan, Nicolette de Keizer, Neill K. J. Adhikari.

**Writing – review & editing:** Bharath Kumar Tirupakuzhi Vijayaraghavan, Ena Gupta, Nagarajan Ramakrishnan, Abi Beane, Rashan Haniffa, Nazir Lone, Nicolette de Keizer, Neill K. J. Adhikari.

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
