## [Decision Letter · Decision Letter 0]

15 Dec 2021

PONE-D-21-24699Barriers and facilitators to the conduct of critical care research in Low and Lower-middle income countries: A scoping reviewPLOS ONE

Dear Bharath Kumar Vijayaraghavan,

Thank you for submitting your manuscript to PLOS ONE. After careful consideration, we feel that it has merit but does not fully meet PLOS ONE’s publication criteria as it currently stands. Therefore, we invite you to submit a revised version of the manuscript that addresses the points raised during the review process.

Please review the methodology of the review. For example quality assessment tools are reported as appropriate in the abstract yet they are not. In addition, the conclusion is rather vague. Instead of stating that the review highlights important barriers, it would be helpful to highlight a key finding as well as implications for practice or research. Do refer to the methodology guidance for scoping reviews and comments given by the editor and peer reviewer below.

We look forward to receiving your revised manuscript.

Kind regards,

Eleanor Ochodo

Academic Editor

PLOS ONE

“Wellcome Trust, U.K. (grant number WT215522/Z19/Z). The funder had no role in the design, conduct, analysis of this scoping review or in the decision to submit for publication”

“Authors RH and AB were co-applicants on the grant.

Wellcome Trust, U.K. (grant number WT215522/Z19/Z). https://wellcome.org/

The funder had no role in the design, conduct, analysis of this scoping review or in the decision to submit for publication”

“No competing interests”

5. Please amend your list of authors on the manuscript to ensure that each author is linked to an affiliation. Authors’ affiliations should reflect the institution where the work was done (if authors moved subsequently, you can also list the new affiliation stating “current affiliation:….” as necessary).

6. We note that this manuscript is a systematic review or meta-analysis; our author guidelines therefore require that you use PRISMA guidance to help improve reporting quality of this type of study. Please upload copies of the completed PRISMA checklist as Supporting Information with a file name “PRISMA checklist”.

**Additional Editor Comments :**

Please consider using the Mixed Method Appraisal Tool (MMAT) for quality assessment or other appropriate tools as advised by the peer reviewer.

A scoping review highlights gaps to be addressed. To help readers better, please include a paragraph on implications for practice and research based on the scoping exercise.

Do refer to the following guidance on scoping reviews:

1. Peters MDJ, Marnie C, Tricco AC, Pollock D, Munn Z, Alexander L, McInerney P, Godfrey CM, Khalil H. Updated methodological guidance for the conduct of scoping reviews. JBI Evid Synth. 2020 Oct;18(10):2119-2126. doi: 10.11124/JBIES-20-00167

2. Peters MD, Godfrey CM, Khalil H, McInerney P, Parker D, Soares CB. Guidance for conducting systematic scoping reviews. Int J Evid Based Healthc. 2015 Sep;13(3):141-6. doi: 10.1097/XEB.0000000000000050

3. https://jbi-global-wiki.refined.site/space/MANUAL/3283910770/Chapter+11%3A+Scoping+reviews

**Reviewers' comments:**

Reviewer's Responses to Questions

**Comments to the Author**

1. Is the manuscript technically sound, and do the data support the conclusions?

Reviewer #1: Yes

Reviewer #2: Partly

2. Has the statistical analysis been performed appropriately and rigorously? 

Reviewer #1: Yes

Reviewer #2: N/A

3. Have the authors made all data underlying the findings in their manuscript fully available?

Reviewer #1: Yes

Reviewer #2: Yes

4. Is the manuscript presented in an intelligible fashion and written in standard English?

Reviewer #1: Yes

Reviewer #2: Yes

5. Review Comments to the Author

Reviewer #1: This is a well executed and written scoping review. The authors have reported their findings in accordance with the PRISMA-ScR checklist. Extending the inclusion criteria to other languages such as French, Spanish and Portuguese may have provided more studies for consideration. However, this may not be a feasible undertaking for the authors.

Reviewer #2: Summary of the research

The authors have summarized the gaps in the health care systems based in low and lower middle-income countries. One of the main concerns is that there is lack of resources that go into research and infrastructure to support the health care systems. They have also noted that high quality research contributes to quality health care system yet this lacks in low and lower middle-income countries where their health systems are wanting. Their focus is on critical illness where contextual issues matter in that evidence generated from high income settings might not be applicable to these settings. Moreover, the authors have highlighted that there is paucity of data regarding critical care research. Therefore, the authors have set out to conduct a scoping review with an aim to describe the barriers and facilitators for the conduct of critical care research in low and lower middle-income countries.

Strengths and limitations of the study

The manuscript has been written clearly. The authors have used standardized and clear methodology. The authors went ahead and conducted quality assessment of the included studies. The authors have also acknowledged their limitations in their methods.

The authors are addressing a key area in health by conducting a scoping review though, they only conducted the search in two databases without providing a justification for their choice, this limits the literature available to provide the required evidence. The eligibility criteria were not very explicit for example type of original studies to be included and the specific study designs. Furthermore, the authors used PRISMA 2009 guidelines to guide their PRISMA flow diagram yet there is an updated PRISMA 2020 guidelines (Page et al,2021).

Examples and Evidence

Major issues

Line 126 to line 132, describes the eligibility criteria for the studies. The authors might want to look at the five included studies again (Table 1) and verify if they meet the eligibility criteria.

1. Andre-von Arnim et al focuses on critical care research in children

2. Sawe et al addresses the barriers and facilitators in implementing the trauma registries

3. Franzen et al addresses the challenges and enablers in conducting clinical trials but the focus is not on critically ill patients

4. Ahmed et al focuses on broader issues of researchers and not the ones involved in care of critically ill patients

5. Alusio et al appears to be a literature review on clinical emergency care research

The authors could come up with a clear eligibility criterion guided by the Population-Concept-Context (PCC) framework designed by the Joanna Briggs Institute for scoping reviews. This will guide them in developing a focused question and in study selection.

Minor issues

1. In line 115, “……. perform an updated review of literature”, this may not be appropriate given the two different methodologies and focus, that the former evidence synthesis Alemayehu et al, was a systematic review on a different focus on clinical trials while for the authors’ synthesis, it is a scoping review with a focus on critically ill patients.

2. In line 49, ‘….to understand barriers and facilitators ….’ while in line 115, ‘….to describe the barriers and opportunities….’. The authors have used different words to state their objective. From the findings of the scoping review ‘….to describe.’ would be more appropriate and scientifically measurable as compared to understand.

3. In line 120, ‘…Ovid versions of MEDLINE and EMBASE.’ The authors searched only two databases. Can the authors provide the justification for using only the two and not any other databases? Given the topic they are addressing, they might have benefited in conducting searches in more than two databases.

Additionally, they can also provide the rationale for restricting the publications to English language only.

4. In line 1,2,101,127,135,191,197,201,246,248,261,265,274 there is use of the abbreviations LMICs and LLMICs. The authors’ focus is on low and lower middle-income countries as indicated in the title. The authors should use the right abbreviation and be consistent throughout the manuscript to prevent confusion to the readers

5. In line 144, the authors indicated they used Cochrane Risk of Bias 2.0 tool to assess quality in the randomized controlled trials (RCTs) yet in the included studies RCTs, were not part of the studies.

6. In line 152, the authors have reported ‘We also provide a descriptive summary of the quality of the included studies.’ However, there is no reference to where the description can be found such as a table, supplementary material etc.

7. In line 182, Table 1: the authors can include year of publication in column one in addition to author and location. For column 2, I would suggest the authors to use the title ‘participants’ so that to eliminate repetition of the word in all the rows. Column 3, ‘type/methods’ title is not clear you need to state … ‘type of…or methods of….’ Or you can use the term ‘study design’ for the title of column 3.

8. In line 197 and 201, table 2 and 3, the authors have reported on the barriers and facilitators for the conduct of critical care research. The authors had indicated they will use themes in data synthesis; however, they have listed the themes from line 188 to line 195 but no further description of the themes.

9. In figure 1, PRISMA flow diagram, the authors have used PRISMA 2009, they should use the updated PRISMA 2020 guidelines (Page et al ,2021) to guide their flow diagram. The authors could add some footnotes to clarify wrong population for instance, is it people who are not critically ill, or participants are from high income settings etc.

Also, wrong study design needs to be clarified since the authors had indicated in line 126-127, that they will include original studies that had used qualitative or quantitative or mixed methods design.

The included studies in last box, it is indicated ‘studies included in qualitative synthesis’, yet this is a scoping review (had both quantitative and qualitative). The authors can revise to read ‘studies included in the synthesis.’

10. In line 272-275, the conclusion is the same as the one in the abstract. In the main manuscript, the conclusion can be revised with focus on the summarized findings in relation to the objective of the review and in addition, the authors can highlight the gap in their findings and recommend the type of research to address it.

11. In S2 appendix, please indicate the tool used for quality assessments in the titles.

12. In S3 appendix, please indicate the study that was assessed using the tool mentioned

13. In line 352, Aluisio AR paper appears to be a literature review on the topic. The authors might want to look at their eligibility criteria which they had stated they will include original studies; therefore, this study might not be included. Nonetheless, the authors might want to look at the references and see if they could identify a study that answers their questions

14. The authors can report on any deviations from the protocol published in the open science framework.

15. Since it is more than six months from their last search (March 2021), the authors could consider doing an updated search.

6. PLOS authors have the option to publish the peer review history of their article (what does this mean?). If published, this will include your full peer review and any attached files.

Reviewer #1: No

Reviewer #2: No

---

## [Author Response · Author response to Decision Letter 0]

11 Feb 2022

We would like to begin by thanking the Editorial team and the Reviewers for the in-depth comments and feedback. We appreciate their insights and feel it has helped improve our manuscript. We respond below (reviewer comments are italicized). 

Additional Editor Comments:

Please consider using the Mixed Method Appraisal Tool (MMAT) for quality assessment or other appropriate tools as advised by the peer reviewer.

Response: Thank you for pointing us to this resource. For the papers identified by us that were eligible for quality assessment ( one of them a quantitative survey and the other four that used qualitative methods), we have used well validated tools (the tools developed by the CLARITY research group for all non-randomized designs, and criteria proposed by Kuper for qualitative studies). We are happy to use the MMAT if the editorial team strongly feels this is essential. 

A scoping review highlights gaps to be addressed. To help readers better, please include a paragraph on implications for practice and research based on the scoping exercise.

Response: We have now added a paragraph in the Discussion section on implications(below). 

“Our scoping review throws light on some of the persistent challenges to the strengthening of research systems in LMICs. It identifies deep-rooted issues that have plagued healthcare systems in resource-constrained settings. Yet, the review also calls attention to potential solutions and opportunities, several of which are immediately feasible and implementable. Some of these relate to greater collaboration between HIC and LMIC researchers, including joint funding applications, development of a strong mentorship network within LMICs as well as between LMIC and HIC researchers, and the sharing of resources (equipment, technology etc.) between these regions. Additional solutions include training clinicians in research methods as well as in grant-writing and manuscript writing, improving the quality of medical documentation and record keeping, and enhancing access to low-cost ubiquitous technology such as mobile phones and tablet-devices for easier data collection and entry. Other solutions specific to critical care research include the development of core dataset for specific disciplines and choosing outcomes that are both relevant, context-specific and feasible. 

Encouragingly, several newer models are addressing some of the above challenges- the development of critical care registries in Asia and Africa focused on evaluating case-mix and outcomes from critical illness with harmonized data collection tools [29,30], the embedding of the Randomized Embedded Multifactorial Adaptive Platform Trial for Community Acquired Pneumonia (REMAP-CAP) within these registries [31], the recently published international collaborative trial comparing two doses of corticosteroids in severe COVID-19 with nearly 40% of trial participants enrolled from India [32] the World Health Organization led pragmatic SOLIDARITY trial [33] are some important examples. “

Do refer to the following guidance on scoping reviews:

1. Peters MDJ, Marnie C, Tricco AC, Pollock D, Munn Z, Alexander L, McInerney P, Godfrey CM, Khalil H. Updated methodological guidance for the conduct of scoping reviews. JBI Evid Synth. 2020 Oct;18(10):2119-2126. doi: 10.11124/JBIES-20-00167

2. Peters MD, Godfrey CM, Khalil H, McInerney P, Parker D, Soares CB. Guidance for conducting systematic scoping reviews. Int J Evid Based Healthc. 2015 Sep;13(3):141-6. doi: 10.1097/XEB.0000000000000050

3. https://jbi-global-wiki.refined.site/space/MANUAL/3283910770/Chapter+11%3A+Scoping+reviews

Response: Thank you for sharing these excellent resources. In referring to these papers, we note that we have followed the guidance provided in them as well in the original 2005 paper by Arksey and O’Malley paper (https://www.tandfonline.com/doi/abs/10.1080/1364557032000119616).

Specifically, we are well aligned with some of the important points of the updated methodological guidance for the conduct of scoping reviews referred by you (doi: 10.11124/JBIES-20-00167). For instance:

1. Definition of scoping review as provided in the paper: “exploratory projects that systematically map the literature available on a topic, identifying key concepts, theories, source of evidence and gaps in the literature”- we believe our work fits well with this definition. We have now addressed the specific lacunae you have highlighted on the need to add a section on implications for practice and research. 

2. Need for a prespecified protocol: “authors conducting a scoping review should consider publishing, registering or making their protocols available via platforms such as Figshare, Open Science Framework..”. Our protocol was registered on Open Science Framework prior to data extraction/analysis. 

3. Need for scoping review to be broad-based, exploratory and focused on maximising the output on knowledge gaps: Keeping this in mind, we shifted our strategy from a restrictive approach to including studies to maximising insights on barriers and facilitators to the conduct of research (e.g. inclusion of Alusio et al). 

We have also broadly adhered to the recommendations and guidance provided in these papers and have also provided the specific PRISMA Scoping review checklist. 

Reviewer #1: 

This is a well-executed and written scoping review. The authors have reported their findings in accordance with the PRISMA-ScR checklist. Extending the inclusion criteria to other languages such as French, Spanish and Portuguese may have provided more studies for consideration. However, this may not be a feasible undertaking for the authors.

Response: Thank you so much for this positive feedback. Yes, inclusion of other languages was not feasible for us, given that we don’t have expertise in any of these languages and our lack of access to resources for translation of such work for review. 

Reviewer #2: 

Summary of the research

The authors have summarized the gaps in the health care systems based in low and lower middle-income countries. One of the main concerns is that there is lack of resources that go into research and infrastructure to support the health care systems. They have also noted that high quality research contributes to quality health care system yet this lacks in low and lower middle-income countries where their health systems are wanting. Their focus is on critical illness where contextual issues matter in that evidence generated from high income settings might not be applicable to these settings. Moreover, the authors have highlighted that there is paucity of data regarding critical care research. Therefore, the authors have set out to conduct a scoping review with an aim to describe the barriers and facilitators for the conduct of critical care research in low and lower middle-income countries.

Strengths and limitations of the study

The manuscript has been written clearly. The authors have used standardized and clear methodology. The authors went ahead and conducted quality assessment of the included studies. The authors have also acknowledged their limitations in their methods.

The authors are addressing a key area in health by conducting a scoping review though, they only conducted the search in two databases without providing a justification for their choice, this limits the literature available to provide the required evidence. The eligibility criteria were not very explicit for example type of original studies to be included and the specific study designs. Furthermore, the authors used PRISMA 2009 guidelines to guide their PRISMA flow diagram yet there is an updated PRISMA 2020 guidelines (Page et al,2021).

Response: We thank the reviewer for this detailed feedback. 

1. Use of two databases only: We restricted our search to the two databases MEDLINE and EMBASE as we only found 6 studies that could be included from these two large and popular databases with robust indexing methods. In discussion between the researchers, it was strongly felt that the additional effort needed to search other databases would be disproportionate to the likely success in finding additional literature. The ‘effort to yield’ ratio was thought to be low and hence we made the decision to stop with the two largest databases. We acknowledge this to be a limitation and have added this as an additional point under limitations. 

2. PRISMA: Sorry for missing the updated version. This is now rectified and the PRISMA 2020 flow diagram has been incorporated. 

Examples and Evidence

Major issues

Line 126 to line 132, describes the eligibility criteria for the studies. The authors might want to look at the five included studies again (Table 1) and verify if they meet the eligibility criteria.

Responses: For the section below, we have included the reviewer text as is and immediately followed it up with a response/justification. 

1. Andre-von Arnim et al focuses on critical care research in children: We did not propose to restrict to adult patients- hence, this study was included. 

2. Sawe et al addresses the barriers and facilitators in implementing the trauma registries: Admissions related to trauma form a major part of intensive care unit admissions globally. It was our view that this would very much be within the purview of the review. 

3. Franzen et al addresses the challenges and enablers in conducting clinical trials but the focus is not on critically ill patients: Yes, we discussed between the 2 reviewers (BKTV and EG) and also with the third reviewer (NJKA)- as the study did not specifically exclude critically ill patients, we proceeded to include. Our search produced limited literature and we were conscious that we don’t exclude any of the relevant published literature in order to ensure that the review maximised the learning opportunities. This approach is consistent with the broad nature of scoping reviews.

We have now added a section under ‘methods’ on ‘deviations from protocol’. 

4. Ahmed et al focuses on broader issues of researchers and not the ones involved in care of critically ill patients: Once again, this paper was included after extensive discussions between the reviewers as it was felt that the paper contributed important information to the challenges of conducting research in LMICs. This paper identifies the challenges faced by a unique cohort of research health professionals with a desire to return to LMICs. This provided an opportunity to examine these responses and recognize these challenges. Hence, we considered this paper valuable to our aims.

5. Alusio et al appears to be a literature review on clinical emergency care research: This paper generated the most debate within our group- however, we finally did include for similar reasons as above- that the paper provided important insights. This paper is a systematic analysis of the focussed barriers to research in emergency care in LMICs, where emergency and critical care are closely linked. 

The authors could come up with a clear eligibility criterion guided by the Population-Concept-Context (PCC) framework designed by the Joanna Briggs Institute for scoping reviews. This will guide them in developing a focused question and in study selection.

Response: Thank you for directing us to this resource. While we did not explicitly use the PCC approach, our study selection was based on a modification of the PICO approach. The corresponding PCC would be:

Population: critically ill patients of any age- we specified in our methods that since the definition of critical care is variable, we included all studies where authors have defined the population as critically ill. The exception to this is the study by Franzen and we have explained above we chose to include this study.

Concept: barriers and facilitators to the conduct of critical care research 

Context: Lower-middle income countries

And the components of this PCC are specified in the section on ‘search strategy and eligibility criteria’ under ‘methods’. 

Despite our prespecified approach (protocol was published apriori) and focused question, the scarcity of literature around the topic of interest meant that we had to make decisions on inclusion on a case-case basis for some of the studies. 

Minor issues

1. In line 115, “……. perform an updated review of literature”, this may not be appropriate given the two different methodologies and focus, that the former evidence synthesis Alemayehu et al, was a systematic review on a different focus on clinical trials while for the authors’ synthesis, it is a scoping review with a focus on critically ill patients.

Response: Thank you- we have modified this statement.

2. In line 49, ‘….to understand barriers and facilitators ….’ while in line 115, ‘….to describe the barriers and opportunities….’. The authors have used different words to state their objective. From the findings of the scoping review ‘….to describe.’ would be more appropriate and scientifically measurable as compared to understand.

Response: Thank you- agree, we have made the change. 

3. In line 120, ‘…Ovid versions of MEDLINE and EMBASE.’ The authors searched only two databases. Can the authors provide the justification for using only the two and not any other databases? Given the topic they are addressing, they might have benefited in conducting searches in more than two databases.

Response: We restricted our search to the two databases MEDLINE and EMBASE as we only found 6 studies that could be included from these two large and popular databases with robust indexing methods. In discussion between the researchers, it was strongly felt that the additional effort needed to search several other databases would be disproportionate to the likely success in finding additional literature. The ‘effort to yield’ ratio was thought to be low and hence we made the decision to stop with the two largest databases. We acknowledge this to be a limitation and have added this as an additional point under limitations. 

Additionally, they can also provide the rationale for restricting the publications to English language only.

Response: Our own limited familiarity with other languages and the lack of resources to search and translate from other languages. We have now added a line under ‘ Search Strategy and eligibility criteria for studies’ to reflect this. 

4. In line 1,2,101,127,135,191,197,201,246,248,261,265,274 there is use of the abbreviations LMICs and LLMICs. The authors’ focus is on low and lower middle-income countries as indicated in the title. The authors should use the right abbreviation and be consistent throughout the manuscript to prevent confusion to the readers

Response: Thank you- agreed and apologise for the varying abbreviations. We have changed this to LMICs through the document for consistency. 

5. In line 144, the authors indicated they used Cochrane Risk of Bias 2.0 tool to assess quality in the randomized controlled trials (RCTs) yet in the included studies RCTs, were not part of the studies.

Response: The intention was to use the ROB 2.0 tool in the event of identifying RCTs. This was prespecified in the methods. Eventually, we did not find any RCTs.

6. In line 152, the authors have reported ‘We also provide a descriptive summary of the quality of the included studies.’ However, there is no reference to where the description can be found such as a table, supplementary material etc.

Response: Apologies for poorly framing the sentence – what we meant is that we have appraised the quality of the included studies. However, this is already mentioned in the previous paragraph under the section ‘study selection, data extraction and quality assessment’. Hence, we have now deleted this sentence as it does not add anything further. 

7. In line 182, Table 1: the authors can include year of publication in column one in addition to author and location. For column 2, I would suggest the authors to use the title ‘participants’ so that to eliminate repetition of the word in all the rows. Column 3, ‘type/methods’ title is not clear you need to state … ‘type of…or methods of….’ Or you can use the term ‘study design’ for the title of column 3.

Response: Thank you- all of these changes are now incorporated. 

8. In line 197 and 201, table 2 and 3, the authors have reported on the barriers and facilitators for the conduct of critical care research. The authors had indicated they will use themes in data synthesis; however, they have listed the themes from line 188 to line 195 but no further description of the themes.

Response: Thank you- we used 7 themes – finance, human capacity, ethics, governance and regulatory, research environment, operational, competing demands and others. The themes are listed in the tables (Table 2 and 3) on barriers and facilitators. We did not provide any additional text as we feel they are self-explanatory. 

9. In figure 1, PRISMA flow diagram, the authors have used PRISMA 2009, they should use the updated PRISMA 2020 guidelines (Page et al ,2021) to guide their flow diagram. The authors could add some footnotes to clarify wrong population for instance, is it people who are not critically ill, or participants are from high income settings etc.

Also, wrong study design needs to be clarified since the authors had indicated in line 126-127, that they will include original studies that had used qualitative or quantitative or mixed methods design.

Response: Thank you- these are now done and the PRISMA 2020 version of the flow diagram has been incorporated. 

The included studies in last box, it is indicated ‘studies included in qualitative synthesis’, yet this is a scoping review (had both quantitative and qualitative). The authors can revise to read ‘studies included in the synthesis.’

Response: Thank you- this is done. 

10. In line 272-275, the conclusion is the same as the one in the abstract. In the main manuscript, the conclusion can be revised with focus on the summarized findings in relation to the objective of the review and in addition, the authors can highlight the gap in their findings and recommend the type of research to address it.

Response: We have now updated the conclusion in the manuscript. 

Below the edited conclusion:

“Our scoping review highlights important and persistent barriers to the conduct of critical care research in LMICs, identifies potential solutions, and informs researchers, policy-makers and governments on the steps necessary for strengthening research systems. While there have been recent encouraging examples that address some of these challenges, broader, multifaceted and systematic strategies with short and longer term goals are essential from Ministries of Health, public health agencies and other key stakeholders to addressing the deep-rooted problems that have plagued research in LMICs. “

11. In S2 appendix, please indicate the tool used for quality assessments in the titles.

Response: Done and also referenced in the main manuscript.

12. In S3 appendix, please indicate the study that was assessed using the tool mentioned

Response: Done and also referenced in the main manuscript. 

13. In line 352, Aluisio AR paper appears to be a literature review on the topic. The authors might want to look at their eligibility criteria which they had stated they will include original studies; therefore, this study might not be included. Nonetheless, the authors might want to look at the references and see if they could identify a study that answers their questions

Response: Thank you- this paper generated the most debate within our group- however, we finally did include for similar reasons as above- that the paper provided important insights. 

We have checked the bibliography of all the included papers. 

14. The authors can report on any deviations from the protocol published in the open science framework.

Response: We have added a section under ‘methods’. 

15. Since it is more than six months from their last search (March 2021), the authors could consider doing an updated search.

Response: This is now done and one additional study Johnson et al. was identified. The manuscript has been updated accordingly. 

Response: These are done. 

Response: Done

“Wellcome Trust, U.K. (grant number WT215522/Z19/Z). The funder had no role in the design, conduct, analysis of this scoping review or in the decision to submit for publication”

“Authors RH and AB were co-applicants on the grant.

Wellcome Trust, U.K. (grant number WT215522/Z19/Z). https://wellcome.org/

The funder had no role in the design, conduct, analysis of this scoping review or in the decision to submit for publication”

Response: This is done now

“No competing interests”

Response: This is done now

5. Please amend your list of authors on the manuscript to ensure that each author is linked to an affiliation. Authors’ affiliations should reflect the institution where the work was done (if authors moved subsequently, you can also list the new affiliation stating “current affiliation:….” as necessary).

Response: This is done

6. We note that this manuscript is a systematic review or meta-analysis; our author guidelines therefore require that you use PRISMA guidance to help improve reporting quality of this type of study. Please upload copies of the completed PRISMA checklist as Supporting Information with a file name “PRISMA checklist”.

Response: The PRISMA checklist specific for Scoping reviews is part of the supplementary appendix.

---

## [Decision Letter · Decision Letter 1]

21 Mar 2022

PONE-D-21-24699R1Barriers and facilitators to the conduct of critical care research in Low and Lower-middle income countries: A scoping reviewPLOS ONE

Dear Bharath Vijayaraghavan,

Thank you for submitting your manuscript to PLOS ONE. After careful consideration, we feel that it has merit but does not fully meet PLOS ONE’s publication criteria as it currently stands. Therefore, we invite you to submit a revised version of the manuscript that addresses the points raised during the review process. Please address the additional minor comments brought forward by reviewer #2. Also do provide some references for the implications of practice section from lines 272 to 278. There are a number of published papers that have evaluated the solutions proposed by the authors. Finally do conduct a spell and grammar check before submission because PLOS ONE does limited copy editing of accepted manuscripts.

Kind regards,

Eleanor Ochodo

Academic Editor

PLOS ONE

Journal Requirements:

Additional Editor Comments:

Please provide some references for the implications of practice section especially from line 272-278. There are already a number of published studies and systematic reviews on the solutions proposed by the author team.

Reviewers' comments:

Reviewer's Responses to Questions

**Comments to the Author**

1. If the authors have adequately addressed your comments raised in a previous round of review and you feel that this manuscript is now acceptable for publication, you may indicate that here to bypass the “Comments to the Author” section, enter your conflict of interest statement in the “Confidential to Editor” section, and submit your "Accept" recommendation.

Reviewer #1: All comments have been addressed

Reviewer #2: (No Response)

2. Is the manuscript technically sound, and do the data support the conclusions?

Reviewer #1: (No Response)

Reviewer #2: Yes

3. Has the statistical analysis been performed appropriately and rigorously? 

Reviewer #1: (No Response)

Reviewer #2: Yes

4. Have the authors made all data underlying the findings in their manuscript fully available?

Reviewer #1: (No Response)

Reviewer #2: Yes

5. Is the manuscript presented in an intelligible fashion and written in standard English?

Reviewer #1: (No Response)

Reviewer #2: Yes

6. Review Comments to the Author

Reviewer #1: (No Response)

Reviewer #2: Summary

The authors have responded comprehensively to the comments and incorporated the changes in the manuscript.

Minor issues

1. In line 134, the context can be modified to read ‘Low and Lower-middle income countries’.

2. In line 147 to 148, ‘Study quality was assessed using Cochrane Risk of Bias 2.0 tool…….’

I suggest this sentence should not be included since there was no RCTs in the included studies but relevant to be included in the protocol.

3. In line 149 to line 150, please check the font size

4. In line 196, Please include year of publication in table 1. It helps the reader to interpret the findings appropriately based on the time the research was conducted. It also saves the reader the time and energy taken to go through the list of references to check the year of publication.

5. Figure 1 is not clear; the words are faint therefore the reader may strain to make sense of it.

7. PLOS authors have the option to publish the peer review history of their article (what does this mean?). If published, this will include your full peer review and any attached files.

Reviewer #1: No

Reviewer #2: No

---

## [Author Response · Author response to Decision Letter 1]

28 Mar 2022

Response to Reviewer Comments

We thank the Editor and Reviewers for the thoughtful feedback and are grateful for their assistance in improving the manuscript. We respond below- all editorial and reviewer comments are italicized with responses below each comment. 

Additional Editor Comments:

Please provide some references for the implications of practice section especially from line 272-278. There are already a number of published studies and systematic reviews on the solutions proposed by the author team.

Response: This is now done- we have added references 29-34 which support the statements in the ‘implications for practice’ section

Reviewer 2 Comments:

Summary:

The authors have responded comprehensively to the comments and incorporated the changes in the manuscript.

Response: Thank you.

Minor issues

1. In line 134, the context can be modified to read ‘Low and Lower-middle income countries’.

Response: this is now done. 

2. In line 147 to 148, ‘Study quality was assessed using Cochrane Risk of Bias 2.0 tool…….’

I suggest this sentence should not be included since there was no RCTs in the included studies but relevant to be included in the protocol.

Response: This sentence has now been deleted.

3. In line 149 to line 150, please check the font size

Response: thank you- corrected. 

4. In line 196, Please include year of publication in table 1. It helps the reader to interpret the findings appropriately based on the time the research was conducted. It also saves the reader the time and energy taken to go through the list of references to check the year of publication.

Response: Thank you- agree and added.

5. Figure 1 is not clear; the words are faint therefore the reader may strain to make sense of it.

Response: We have now revised this figure for better clarity.

---

## [Editor Report · Decision Letter 2]

29 Mar 2022

Barriers and facilitators to the conduct of critical care research in Low and Lower-middle income countries: A scoping review

PONE-D-21-24699R2

Dear Bharath Vijayaraghavan,

We’re pleased to inform you that your manuscript has been judged scientifically suitable for publication and will be formally accepted for publication once it meets all outstanding technical requirements.

Kind regards,

Eleanor Ochodo

Academic Editor

PLOS ONE

Additional Editor Comments:

Table 1 will need to be reformatted or resized to fit to the page.

---

## [Editor Report · Acceptance letter]

31 Mar 2022

PONE-D-21-24699R2 

Barriers and facilitators to the conduct of critical care research in Low and Lower-middle income countries: A scoping review 

Dear Dr. Tirupakuzhi Vijayaraghavan:

I'm pleased to inform you that your manuscript has been deemed suitable for publication in PLOS ONE. Congratulations! Your manuscript is now with our production department. 

Kind regards, 

on behalf of

Prof Eleanor Ochodo 

Academic Editor

PLOS ONE